# Evaluation of Rosmarinic Acid on Broiler Growth Performance, Serum Biochemistry, Liver Antioxidant Activity, and Muscle Tissue Composition

**DOI:** 10.3390/ani12233313

**Published:** 2022-11-27

**Authors:** Rongsheng Shang, Lifei Chen, Yizhen Xin, Guiying Wang, Rong Li, Shaojin Li, Lusheng Li

**Affiliations:** 1School of Agricultural Science and Engineering, Liaocheng University, Liaocheng 252000, China; 2Shandong Province Engineering Research Center of Black Soldier Fly Breeding and Organic Waste Conversion, Liaocheng University, Liaocheng 252000, China

**Keywords:** rosmarinic acid, broilers, growth performance, serum biochemical indices, antioxidant activity, muscle composition

## Abstract

**Simple Summary:**

Rosmarinic acid (RA) is a natural water-soluble phenolic compound and a major antioxidant occurring in rosemary. In this study, the effects of dietary supplementation with RA on the growth performance, serum biochemical indices, liver antioxidant activity, and muscle nutritional components of broiler chickens were examined. The results showed that supplementation with RA significantly improved the growth performance of broilers. Therefore, the addition of RA to broiler feed as a natural antioxidant has great prospects.

**Abstract:**

The aim of the present study was to investigate the effects of dietary supplementation with different doses of rosmarinic acid (RA) on the growth performance, serum biochemical indices, liver antioxidant activity, and muscle nutritional component of broiler chickens during 1–20 days of age. A total of 1000 1-day-old Cobb 500 white-feathered broilers were divided into five groups. Each group included four replicates and 50 birds per replicate. The control group was the basic fodder group fed with a basal diet. The experimental groups included four levels of RA (100, 200, 300, and 400 mg/kg RA added to the basal diet, respectively). The results showed that a quadratic increase in final body weight (BW) and average daily gain (ADG) and a quadratic decrease in the feed/gain (F/G) ratio were observed with increasing dietary RA levels. At 200 mg/kg RA supplementation, growth performance was significantly improved. Moreover, under this RA treatment, the highest levels of total protein and globulin were detected in the serum, the activities of total superoxide dismutase and catalase in the liver of broilers were significantly increased, and the malondialdehyde content was significantly decreased, indicating a higher antioxidant capacity of the liver when 200 mg/kg RA was added to the basal diet. The flavor of the muscle meat was improved by dietary supplementation with 200 mg/kg RA. Therefore, adding 200 mg/kg of RA to the diet could significantly improve the growth performance of broilers, improve liver antioxidant capacity, and improve muscle quality, etc. The addition of RA to broiler feed as a natural antioxidant has great prospects.

## 1. Introduction

Redox reactions are important biochemical processes in animal cells [1] and, under healthy conditions, oxidation and antioxidant systems are in a dynamic balance. However, under stress, the production of reactive oxygen species (ROS) may exceed the scavenging capacity of antioxidants, thus causing oxidative stress, entailing damage to cell macromolecules, cell damage, and death, and adverse effects on the functioning and survival of the entire organism [2,3]. ROS may originate from endogenous and exogenous sources [4]. In production animals, high-density rearing, intensive feeding, and environmental factors such as temperature changes, hypoxia, and transportation can cause oxidative stress and other adverse physiological consequences which may lead to a reduced production performance [1,5,6]. Livestock and poultry are frequently infected with pathogenic bacteria or viruses, which cause oxidative stress and multiple organ damage, thereby compromising production performance [7,8]. In addition, toxins produced during the storage of raw feed materials can induce oxidative stress, resulting in liver and kidney damage and associated afflictions [9,10]. Oxidative stress caused by external factors such as high temperature and hypoxia in a high-density feeding environment is inevitable during the rearing period. Therefore, supplementing natural antioxidants in the diet of producing animals may help to prevent this adverse process, so as to further improve the growth performance, antioxidant capacity, and muscle quality of these animals.

Rosmarinic acid (RA; molecular formula C_18_H_16_O_8_) is a natural water-soluble phenolic compound with physiological effects and can be isolated from rosemary (*Rosmarinus officinalis*) and plants of the genera *Perilla* and *Prunella* [11,12]. RA is a major antioxidant occurring in rosemary [13]. Compared with other natural antioxidants, RA has a strong scavenging activity against free radicals in vivo, and its antioxidant activity is stronger than that of caffeic acid, chlorogenic acid, and folic acid [12,14]. In addition, RA has anti-inflammatory, antibacterial, antiviral, and immune-regulatory functions [15]. RA is thus commonly used in foods, medicines, and cosmetics [16]. In recent years, the application of rosemary extract or rosemary essential oil as an additive in livestock and poultry rearing has been a hot research topic. Norouzi et al. reported the effect of different dietary levels of rosemary herb powders on the growth performance, carcass traits, and ileal micro-biota of broilers [17]. Ghozlan et al. study showed that adding high concentrations of rosemary to the diet had no significant effect on the growth performance of broilers, and appropriate concentrations of rosemary increased the immunity and antioxidant activities of broilers [18]. The addition of rosemary essential oils improved some production performance and antioxidant enzyme activity (glutathione peroxidase) in heat-stressed laying hens [19]. Dietary supplementation with rosemary (0.5%) was more effective in improving aflatoxin B1-contaminated tilapia feed and enhancing fish immunity [20]. However, studies on the antioxidant, anti-inflammatory, and antiviral properties of RA as the main component of rosemary have mainly focused on rats [21,22,23,24,25]. Whether adding RA to the diet has the above functions in livestock and poultry rearing deserves further study. This experiment was conducted to observe the effects of different doses of RA on growth performance, serum biochemical indices, antioxidant capacity, and proximate and amino acid (AA) compositions of the muscle of broilers, in order to provide a theoretical basis for the application of RA in broiler production.

## 2. Materials and Methods

### 2.1. Study Animals and Experimental Design

The animal experiment was approved by the Institutional Animal Care and Use Committee of Liaocheng University, Liaocheng, China. All experimental procedures involving the use of animals were conducted in compliance with the relevant laws and institutional guidelines. The experiment was conducted at a commercial farm (Shandong woneng Agricultural Technology Co., Ltd., Liaocheng, China) approximately 8 km east of Liaocheng city, Shandong Province, Eastern China. One thousand one-day-old Cobb 500 male and female white-feathered broilers (Liaocheng Fengxiang Group Co., Ltd., Liaocheng, China) with similar initial body weights (BWs) (36.594 ± 0.140 g) were randomly selected and assigned to five groups (i.e., a control group and treatment groups RA-1, RA-2, RA-3, and RA-4, respectively), with four replicates in each group, and 50 broilers per replicate. RA solid powder (Hunan Jinhan Pharmaceutical Co. Ltd., Changsha, China) was added to the basal diet at 0, 100, 200, 300, or 400 mg/kg. According to the technical specifications of broiler-feeding tests (GB/T 40942-2021), the nutritional requirements were combined with the actual local situation to prepare a basic diet (corn-soybean meal type). The diet composition and nutritional parameters are shown in Table 1. Broilers were housed in single-layer cages. The initial temperature of the chick house was approximately 35 °C and was then slowly decreased to 22 °C. The chick house was lit 24 h per day and was naturally ventilated. The chicks had free access to food and water; all individuals had been immunized according to a routine immunization program, and their health status and behavior were observed on a daily basis. The experimental period was 20 d.

### 2.2. Sample Collection

On day 20, two physically fit broilers with a body weight near the average of their respective replicate groups were selected from each treatment replicate. Blood samples (20 mL) were taken from the wing vein, then serum was separated by centrifugation at 3000 rpm and 4 °C for 10 min. Serum samples were stored at −80 °C pending analysis. After blood samples were taken, the broilers were slaughtered by exsanguination after overnight fasting, and dissected on ice bags. After the broiler was slaughtered, the left leg muscle was taken and stored at −80 °C pending analysis. The livers were harvested, and their appendages were removed and cleaned with deionized water. Water on the liver surface was dried using filter paper, and 0.86% physiological saline was added at nine-fold the volume for homogenization, followed by centrifugation at 4000 rpm and 4 °C for 20 min. The supernatants were stored at −80 °C.

### 2.3. Analysis of Growth Performance

During the experimental period, the broilers’ feed intake was recorded daily for each replicate group. The broilers were weighed with an empty stomach in the mornings at 1 day of age (initial body weight) and at 20 days of age (final body weight, final BW). The average daily gain (ADG), average daily feed intake (ADFI), and feed-to-gain ratio (F/G) were calculated, and the survival rate was recorded, according to the following equations:Average daily gain (ADG) = (final body weight − initial body weight)/test days(1)
Average daily feed intake (ADFI) = total feed intake/(test days × total number of test broilers)(2)
Feed-to-gain ratio (F/G) = average daily feed intake/average daily gain(3)
Survival rate = number of live broilers/total broilers × 100%(4)

### 2.4. Serum Biochemical Indices and Liver Antioxidant Enzyme Activity

The levels of total protein (TP), glucose (GLU), globulin (GLOB), albumin (ALB), triglycerides (TGs), cholesterol (CHOL), and urea nitrogen (UN), and the activities of alkaline phosphatase (ALP), aspartate aminotransferase (AST), and alanine aminotransferase (ALT) in the serum were determined using respective commercially available kits (Shanghai Enzyme-linked Biotechnology Co., Ltd., Shanghai, China). The catalog numbers of the kits corresponding to the above determination parameters are ml-094997, ml-095040, ml-060789, ml-095005, ml-095104, ml-094952, ml-076478, ml-095274, ml-095194, and ml-095162, respectively. The total antioxidant capacity (T-AOC) and malondialdehyde (MDA) content, activities of glutathione peroxidase (GSH-PX), catalase (CAT), and total superoxide dismutase (T-SOD) in the liver were also determined using commercial kits (Shanghai Enzyme-linked Biotechnology Co., Ltd., Shanghai, China). The catalog numbers of the kits corresponding to the above determination parameters are ml-E11386, ml-E11338, ml-E11346, ml-E11366, and ml-E11364, respectively.

### 2.5. Analysis of Proximate and AA Compositions of Muscle Tissue

The proximate and AA compositions of the diet and left leg muscle tissue of broilers were determined as previously described [26,27]. In short, the contents of crude protein and crude fat were measured by the Kjeldahl method and Soxhlet extraction method, respectively. After heating the sample to 550 °C in a muffle furnace for 4 h, it was weighed and the ash content determined. After placing the sample in an oven at 105 °C to a constant weight, the sample was cooled and dried in a dryer to measure the moisture content. AA compositions were assessed as previously reported [28]. Briefly, the samples were hydrolyzed using 6 mol/L hydrochloric acid at 110 °C under a nitrogen atmosphere for 22 h, and the AA composition was determined using an Amino Acid Analysis System (Hitachi L-8900; Agilent Technologies, Inc., Santa Clara, CA, USA). The levels of essential amino acids such as methionine (Met), phenylalanine (Phe), leucine (Leu), threonine (Thr), lysine (Lys), isoleucine (Ile), and valine (Val), and non-essential amino acids such as histidine (His), glycine (Gly), aspartic acid (Asp), tyrosine (Tyr), alanine (Ala), glutamic acid (Glu), arginine (Arg), serine (Ser), and proline (Pro) were determined.

### 2.6. Statistical Analyses

All data were analyzed statistically by one-way ANOVA and Tukey’s multiple comparison of variance using SPSS software (version 22.0; SPSS Inc., Chicago, IL, USA). Orthogonal polynomials were used to test linear, quadratic, and cubic responses to dietary levels of RA. Values are expressed as means ± SEM (standard error of mean), and statistical significance is reported at *p* < 0.05.

## 3. Results

### 3.1. Growth Performance

The growth performance results are shown in Table 2. There was no significant difference (*p* > 0.05) in the initial weight of each group of broilers, indicating that the selected broilers met the experimental requirements. The final BW, ADG, and F/G of the RA-1, RA-2, RA-3, and RA-4 broilers differed significantly from that of the control group (*p* < 0.05). The final BW and ADG were significantly higher in RA-2, RA-3, and RA-4 broilers than in control and RA-1 (*p* < 0.05). The ADFI was significantly lower in RA-1 than in RA-3 and RA-4 broilers (*p* < 0.05). The F/G was significantly higher in RA-2 than in the control, RA-1, and RA-4 broilers, and it first showed a quadratic decrease (*p* < 0.001) and then increased with increasing dietary RA levels (*p* < 0.001). RA-2 broilers showed the lowest F/G ratio.

### 3.2. Serum Biochemical Indices

Serum biochemical indices are shown in Table 3. ALB, CHOL, TGs, UN, and GLU levels and ALP, ALT, and AST activities in serum did not differ significantly between treatments. TP levels were significantly higher in the RA-2 group than in the control, RA-1, and RA-4 groups (*p* < 0.001), and they first showed a quadratic increase (*p* < 0.001) and then decreased with increasing dietary RA levels (*p* < 0.05). GLOB levels were significantly higher in the RA-2 group than in the control and RA-1 groups (*p* < 0.05), and they first showed a quadratic increase (*p* < 0.001) and then decreased with increasing dietary RA levels (*p* < 0.05).

### 3.3. Liver and Serum Antioxidant Activity

The results of liver antioxidant activity are shown in Table 4. The activity of GSH-Px in the liver was not significantly affected by dietary RA levels. The T-AOC and T-SOD activities first showed a quadratic increase (*p* < 0.001 and *p* < 0.001, respectively) and then decreased with increasing dietary RA levels. The maximum T-AOC and T-SOD activities were observed in the RA-2 group. MDA levels were significantly lower in the RA-2 group than in the control and RA-1 and RA-4 broilers, and they showed a quadratic decrease (*p* < 0.001) and then increased with increasing dietary RA levels. The lowest level of MDA was observed in RA-2 broilers. CAT activity was significantly higher in RA-2 than in control, RA-1, RA-3, and RA-4 broilers, and it first showed a quadratic increase (*p* < 0.001) and then decreased with increasing dietary RA levels. The highest CAT activity in the liver was observed in RA-2 broilers. The results of serum antioxidant activity are shown in Table 5. The activity of T-AOC in the serum was not significantly affected by dietary RA levels. Compared with the control group, GSH-Px, T-SOD, and CAT activities were significantly increased in the RA-2 and RA-3 groups (*p* < 0.005), and the MDA content of the RA-2, RA-3, and RA-4 groups was significantly decreased (*p* < 0.005).

### 3.4. Muscle Tissue Proximate and AA Composition

The proximate and AA compositions of broiler muscle tissue are shown in Table 6 and Table 7, respectively. Moisture, crude protein, crude fat, and ash content in muscle tissue did not differ significantly between treatments, nor did the muscle tissue concentrations of Met, Val, Phe, Ile, Leu, Tyr, Gly, Ala, Asp, Pro, Arg, and His. The Thr content was significantly higher in RA-3 than in control and RA-1 broilers, and it first showed a quadratic increase (*p* = 0.019) and then decreased with increasing dietary RA levels (*p* < 0.05). The highest Thr content in muscle tissue was observed in RA-3 broilers. The Lys content in muscle tissue showed a linear increase (*p* = 0.008) with increasing dietary RA levels. The Glu content was significantly higher in RA-2 than in RA-3 and RA-4 broilers, and it first showed a quadratic increase (*p* = 0.037) and then decreased with increasing dietary RA levels (*p* < 0.05). The highest Glu content in muscle tissue was observed in RA-2 broilers. The Ser content in muscle tissue showed a linear increase (*p* < 0.001) with increasing dietary RA levels, and it was significantly higher in RA-4 than in control, RA-1, and RA-3 broilers.

## 4. Discussion

RA is one of the main component of rosemary extract [29]. Previous studies have shown that RA has antioxidant, anti-inflammatory, antibacterial, and antiviral effects [21,22,23], which has attracted much attention. The results of the present study showed that the growth performance (in terms of final BW, ADFI, and ADG) of broilers treated with RA in the diet was significantly improved compared with the control group. This may be because the addition of RA improved the anti-inflammatory effect in broilers or improved their intestinal flora, thereby improving their growth performance [22,30]. The appropriate addition of RA to diets could significantly improve the growth performance of these animals. Chen et al. reported that adding 30 mg/kg RA to the diet could significantly improve the growth performance of broilers [31]. Ghazalah et al. reported that dietary supplementation of 500 mg/kg rosemary could significantly improve the growth performance of broilers [32]. In the present study, the lowest F/G was obtained when adding 200 mg/kg RA, demonstrating that this proportion of RA was the best for the growth of broilers. This variation in the optimal amount of RA to add to the diet may be caused by different feeding environments and different dietary formulas [33]. Therefore, it is important to supply RA at an adequate dosage in order to effectively improve the growth performance of broilers.

Serum biochemical indices are typically effective indicators for monitoring physiological conditions and responses to new feed ingredients [9,25]. The results of the present study showed that the levels of TP and GLOB in the serum of broilers fed with RA-supplemented diets were higher than those in the control group, indicating that RA supplementation promoted the synthesis of protein in vivo and improved immune functions, thereby promoting growth, which was consistent with the above results regarding growth performance [34,35]. ALT and AST are important amino acid transaminases that are abundant in animal mitochondria, and their activity changes are sensitive indicators of damage to liver and heart cells [36,37]. ALT and AST levels were within the normal respective range in the present study, and no significant difference between treatments was observed, indicating that RA supplementation had no adverse effect on the liver functions of broilers. The reason may be that RA is nontoxic, and it can potentially maintain glycemic control and regulate the key enzymes of carbohydrate metabolism [38]. The broilers were healthy and had normal protein metabolism. The addition of RA to the diet had no significant effect on other serum biochemical indices, indicating that the addition of RA had no negative effect on broiler growth.

The liver is a crucial organ regarding the growth performance of livestock [10]. Liver antioxidant capacity reflects the growth status of livestock, to a certain extent, and the scavenging of free radicals mainly depends on various antioxidant enzymes secreted by autologous cells [39]. For example, T-SOD can scavenge free superoxide anion radicals (O^2−^), GSH-Px can convert hydrogen peroxide (H_2_O_2_) to water and lipid peroxide to alcohol, and CAT can prevent potential harm from hydroxyl radicals by scavenging H_2_O_2_ [40]. Therefore, liver antioxidant enzyme activity is an important manifestation of animal fitness [41]. The results of the present study showed that under dietary supplementation with 200 mg/kg RA, the activities of T-SOD and CAT in the liver of broilers were significantly increased. This may be because RA could reduce oxidative stress by preventing lipid peroxidation and nitric oxide production, as well as restarting the activity of the GPx and SOD enzymes [42]. MDA is a product of ROS that attacks polyunsaturated fatty acids in biological membranes. The MDA content in the liver not only reflects the degree of lipid peroxidation mediated by oxygen free radicals but also indirectly reflects the degree of oxidative damage [43]. The MDA content of liver and serum was significantly decreased in this study. This is in line with the results of previous studies in which rosemary was added to the diet of broilers [18]. The observed effects may be explained by the RA-mediated transmission of the H atom of -OH groups to free radicals, thus forming a stable new compound without antioxidant effects and terminating the free radical reaction chain [44,45].

Moisture, crude protein, and crude fat content directly affect the nutritional value and flavor of meat, and moisture content is negatively correlated with water retention [46]. In the current study, RA supplementation did not affect the content of moisture, crude protein, or crude fat in broiler leg muscle tissue. AA content and composition are important indices that affect the quality of meat and its products [47]. Thr cannot be synthesized and is the third limiting essential amino acid after methionine and lysine in corn-soybean-based diets of broilers [48]. Adding an appropriate amount of Thr into the diet can promote the growth performance of animals and increase the content of GLOB in serum, so as to improve immunity [49,50]. In this study, the Thr content significantly increased when 200 mg/kg RA was added to the diet. This is consistent with the above conclusions regarding the growth performance and increase in serum GLOB content of broilers, indicating that the increase in Thr content played a positive role when the diet was supplemented with 200 mg/kg RA. The content of Lys in the 400 mg/kg RA treatment group increased to a certain extent. No significant changes were observed in the other essential AAs. This indicated that RA had little effect on the structural stability of muscle proteins in broilers, as suggested previously [51]. Regarding flavor AAs, Glu is not only a structural amino acid of protein or peptide but also a free amino acid. It is delicious and can improve the deposition of flavor substances in muscles [52,53]. Ser is a non-essential AA, which is an important precursor involved in the synthesis of intracellular biological substances such as purine, pyrimidine, and phospholipid [54,55]. Compared with the control group, the content of Glu and Ser in the muscle tissue of the leg increased significantly in broilers receiving 200 mg/kg RA. This indicated that muscle quality and flavor had been greatly improved. Therefore, dietary RA supplementation could increase the levels of essential AAs (Lys, Thr) and flavor AAs (Glu, Ser), thereby improving muscle quality and flavor.

## 5. Conclusions

Dietary supplementation with 200 mg/kg RA significantly improved the growth performance of broilers and reduced the F/G ratio; this treatment produced the highest levels of TP and GLOB in the serum. Moreover, the activities of T-SOD and CAT in the liver of broilers were significantly increased and the MDA content was significantly decreased, indicating that the liver had a high antioxidant capacity when 200 mg/kg RA was added to the basal diet. Meanwhile, the supplementation of RA in diets could increase the content of essential AAs (Lys, Thr) and flavor AAs (Glu, Ser), so as to improve muscle quality and flavor. Therefore, an appropriate amount of RA could be added to the diet as an alternative antioxidant, which has broad development prospects in animal husbandry.

## Figures and Tables

**Table 1 animals-12-03313-t001:** Ingredient and nutrient composition of experimental diets.

Items	Content
Ingredient (%)	
Corn	58.0
Soybean meal	21.0
Wheat flour	5.0
Cottonseed protein	3.0
Calcium hydrogen phosphate	1.5
Broiler premix ^1^	3.0
Limestone	1.2
Corn gluten meal	3.0
Soybean oil	4.0
Salt	0.3
Total	100
Nutrient levels ^2^ (%)	
Metabolizable energy, MJ/kg	12.60
Crude protein	19.11
Calcium	0.90
Non-phytate phosphorus	0.45
Lysine	1.13
Methionine	0.43

^1^ Provided per kilogram of diet: Cu 6.8 mg, Fe 66 mg, Zn 83 mg, Mn 80 mg, I 1 mg, Se 0.30 mg, vitamin A 11,700 IU, vitamin D3 3600 IU, vitamin E 21 IU, vitamin K3 4.2 mg, vitamin B1 3 mg, vitamin B2 10.2 mg, vitamin B6 5.4 mg, folic acid 0.9 mg, D-pantothenic acid 15 mg, nicotinic 45 mg, biotin 0.15 mg, vitamin B12 24 μg. ^2^ Nutrient levels were calculated values.

**Table 2 animals-12-03313-t002:** The effect of dietary supplementation with rosmarinic acid (RA) on growth performance of broilers.

Item ^1^	Control	Treatment ^2^				SEM ^3^	*p* Value	Linear	Quadratic	Cubic
		RA-1	RA-2	RA-3	RA-4					
Initial BW (g)	36.740	36.336	36.535	36.911	36.450	0.140	0.750	0.995	0.961	0.191
Final BW (g)	712.500 ^c^	743.750 ^b^	813.750 ^a^	813.750 ^a^	787.500 ^a^	9.822	<0.001	<0.001	<0.001	0.032
ADFI (g/d)	38.952 ^ab^	38.168 ^b^	38.983 ^ab^	39.833 ^a^	39.953 ^a^	0.212	0.026	0.008	0.210	0.069
ADG (g)	33.788 ^c^	35.371 ^b^	38.861 ^a^	38.842 ^a^	37.553 ^a^	0.489	<0.001	<0.001	<0.001	0.032
F/G	1.153 ^a^	1.080 ^b^	1.003 ^d^	1.026 ^cd^	1.065 ^bc^	0.137	<0.001	0.001	<0.001	0.716
Survival rate (%)	97.436	98.718	96.154	98.077	95.513	0.489	0.203	0.185	0.414	0.845

^a,b,c,d^ Means the values within a row with no common letters differ significantly (*p* < 0.05). ^1^ BW = body weight; ADFI = average daily feed intake; ADG = average daily gain; F/G = feed-to-gain ratio. ^2^ RA-1, diet supplemented with 100 mg/kg rosmarinic acid; RA-2, diet supplemented with 200 mg/kg rosmarinic acid; RA-3, diet supplemented with 300 mg/kg rosmarinic acid; RA-4, diet supplemented with 400 mg/kg rosmarinic acid. ^3^ Values are mean ± SEM (*n* = 4).

**Table 3 animals-12-03313-t003:** The effect of dietary supplementation with rosmarinic acid (RA) on serum biochemical indices of broilers.

Item ^1^	Control	Treatment ^2^				SEM ^3^	*p* Value	Linear	Quadratic	Cubic
		RA-1	RA-2	RA-3	RA-4					
TP (g/L)	24.614 ^c^	28.160 ^b^	31.230 ^a^	30.398 ^ab^	28.129 ^b^	0.509	<0.001	0.001	<0.001	0.717
ALB (g/L)	15.063	15.269	14.116	15.341	13.87	0.240	0.158	0.167	0.620	0.420
GLOB (g/L)	9.551 ^c^	12.895 ^bc^	17.114 ^a^	16.160 ^ab^	14.159 ^ab^	0.605	<0.001	<0.001	<0.001	0.552
TGs (mmol/L)	0.486	0.490	0.485	0.478	0.503	0.112	0.973	0.811	0.686	0.622
CHOL (mmol/L)	4.034	4.116	3.955	4.105	4.161	0.542	0.794	0.541	0.583	0.706
UN (mmol/L)	0.609	0.598	0.578	0.591	0.601	0.006	0.544	0.617	0.135	0.906
GLU (mmol/L)	14.640	13.524	14.446	14.279	13.690	0.327	0.798	0.635	0.990	0.310
ALP (U/L)	904.71	879.574	906.099	916.539	881.880	8.062	0.538	0.881	0.610	0.102
ALT (U/L)	10.340	9.520	9.203	11.184	11.440	0.337	0.139	0.100	0.109	0.336
AST (U/L)	193.669	192.509	185.361	188.075	192.293	2.273	0.770	0.667	0.299	0.653

^a,b,c^ Means the values within a row with no common letters differ significantly (*p* < 0.05). ^1^ TP = total serum protein; ALB = albumin; GLOB = globulin; CHOL = cholesterol; TG = triglyceride; UN = urea nitrogen; GLU = glucose; ALP = alkaline phosphatase; ALT = alanine aminotransferase; AST = aspartate aminotransferase. ^2^ RA-1, diet supplemented with 100 mg/kg rosmarinic acid; RA-2, diet supplemented with 200 mg/kg rosmarinic acid; RA-3, diet supplemented with 300 mg/kg rosmarinic acid; RA-4, diet supplemented with 400 mg/kg rosmarinic acid. ^3^ Values are mean ± SEM (*n* = 8).

**Table 4 animals-12-03313-t004:** The effect of dietary supplementation with rosmarinic acid (RA) on liver antioxidant activity of broilers.

Item ^1^	Control	Treatment ^2^				SEM ^3^	*p* Value	Linear	Quadratic	Cubic
		RA-1	RA-2	RA-3	RA-4					
T-AOC (U/mL)	12.848 ^c^	13.529 ^abc^	14.005 ^a^	13.784 ^ab^	13.310 ^bc^	0.105	0.002	0.064	<0.001	0.948
GSH-Px (U/mL)	831.103	850.289	854.714	862.85	860.750	5.089	0.299	0.051	0.361	0.899
MDA (nmol/mL)	2.883 ^a^	2.410 ^b^	1.740 ^c^	2.073 ^bc^	2.251 ^b^	0.083	<0.001	0.001	<0.001	0.918
T-SOD (U/mL)	125.754 ^c^	160.550 ^a^	163.891 ^a^	147.45 ^ab^	131.80 ^bc^	3.454	<0.001	0.957	<0.001	0.088
CAT (U/mL)	1.589 ^c^	2.210 ^b^	2.518 ^a^	2.224 ^b^	2.169 ^b^	0.058	<0.001	<0.001	<0.001	0.022

^a,b,c^ Means the values within a row with no common letters differ significantly (*p* < 0.05). ^1^ T-AOC = total antioxidant capacity; GSHPx = glutathione peroxidase; MDA = malonaldehyde; T-SOD = total superoxide dismutase; CAT = catalase. ^2^ RA-1, diet supplemented with 100 mg/kg rosmarinic acid; RA-2, diet supplemented with 200 mg/kg rosmarinic acid; RA-3, diet supplemented with 300 mg/kg rosmarinic acid; RA-4, diet supplemented with 400 mg/kg rosmarinic acid. ^3^ Values are mean ± SEM (*n* = 8).

**Table 5 animals-12-03313-t005:** The effect of dietary supplementation with rosmarinic acid (RA) on serum antioxidant activity of broilers.

Item ^1^	Control	Treatment ^2^				SEM ^3^	*p* Value	Linear	Quadratic	Cubic
		RA-1	RA-2	RA-3	RA-4					
T-AOC (U/mL)	8.518	9.080	9.415	9.398	9.108	0.115	0.080	0.056	0.028	0.951
GSH-Px (U/mL)	633.233 ^b^	651.663 ^ab^	657.835 ^a^	658.642 ^a^	657.745 ^a^	3.101	0.033	0.008	0.065	0.589
MDA (nmol/mL)	2.655 ^a^	2.150 ^ab^	1.836 ^b^	1.653 ^b^	1.903 ^b^	0.089	0.001	<0.001	0.007	0.611
T-SOD (U/mL)	119.710 ^b^	160.103 ^a^	168.112 ^a^	154.825 ^a^	153.345 ^a^	4.113	<0.001	0.006	<0.001	0.043
CAT (U/mL)	1.487 ^b^	1.882 ^ab^	2.128 ^a^	2.090 ^a^	1.817 ^ab^	0.748	0.036	0.075	0.007	0.854

^a,b,c^ Means the values within a row with no common letters differ significantly (*p* < 0.05). ^1^ T-AOC = total antioxidant capacity; GSHPx = glutathione peroxidase; MDA = malonaldehyde; T-SOD = total superoxide dismutase; CAT = catalase. ^2^ RA-1, diet supplemented with 100 mg/kg rosmarinic acid; RA-2, diet supplemented with 200 mg/kg rosmarinic acid; RA-3, diet supplemented with 300 mg/kg rosmarinic acid; RA-4, diet supplemented with 400 mg/kg rosmarinic acid. ^3^ Values are mean ± SEM (*n* = 8).

**Table 6 animals-12-03313-t006:** The effect of dietary supplementation with rosmarinic acid (RA) on muscle proximate composition of broilers.

Item	Control	Treatment ^1^				SEM ^2^	*p* Value	Linear	Quadratic	Cubic
		RA-1	RA-2	RA-3	RA-4					
Moisture (%)	10.141	10.220	10.251	10.211	10.219	0.034	0.896	0.558	0.470	0.703
Crude protein (%)	78.640	79.048	79.249	78.761	79.110	0.251	0.944	0.726	0.715	0.577
Crude fat (%)	4.310	4.431	4.520	4.561	4.481	0.418	0.378	0.117	0.205	0.765
Ash (%)	5.465	5.511	5.411	5.359	5.331	0.417	0.662	0.172	0.781	0.573

^1^ RA-1, diet supplemented with 100 mg/kg rosmarinic acid; RA-2, diet supplemented with 200 mg/kg rosmarinic acid; RA-3, diet supplemented with 300 mg/kg rosmarinic acid; RA-4, diet supplemented with 400 mg/kg rosmarinic acid. ^2^ Values are mean ± SEM (*n* = 8).

**Table 7 animals-12-03313-t007:** The effect of dietary supplementation with rosmarinic acid (RA) on muscle amino acids of broilers.

Item ^1^	Control	Treatment ^2^				SEM ^3^	*p* Value	Linear	Quadratic	Cubic
		RA-1	RA-2	RA-3	RA-4					
Thr (%)	0.580 ^c^	0.641 ^bc^	0.663 ^ab^	0.711 ^a^	0.675 ^ab^	0.009	<0.001	<0.001	0.019	0.438
Met (%)	0.435	0.439	0.466	0.419	0.463	0.008	0.372	0.563	0.944	0.268
Val (%)	0.781	0.819	0.828	0.815	0.809	0.112	0.810	0.565	0.305	0.694
Lys (%)	1.488 ^b^	1.513 ^ab^	1.539 ^ab^	1.544 ^ab^	1.584 ^a^	0.111	0.110	0.008	0.927	0.675
Phe (%)	0.668	0.683	0.675	0.636	0.660	0.006	0.210	0.184	0.798	0.068
Ile (%)	0.794	0.848	0.863	0.845	0.870	0.115	0.345	0.095	0.391	0.360
Leu (%)	1.360	1.318	1.319	1.356	1.344	0.077	0.343	0.971	0.178	0.122
Glu (%)	2.708 ^ab^	2.731 ^ab^	2.773 ^a^	2.674 ^b^	2.681 ^b^	0.103	0.015	0.110	0.037	0.195
Ser (%)	0.460 ^c^	0.488 ^bc^	0.540 ^ab^	0.500 ^bc^	0.585 ^a^	0.010	<0.001	<0.001	0.738	0.085
Tyr (%)	0.441	0.480	0.453	0.493	0.476	0.007	0.170	0.114	0.485	0.845
Gly (%)	0.750	0.768	0.810	0.758	0.771	0.111	0.601	0.714	0.332	0.642
Ala (%)	0.888	0.916	0.908	0.924	0.904	0.107	0.894	0.634	0.467	0.988
Asp (%)	1.549	1.543	1.579	1.593	1.521	0.176	0.816	0.973	0.382	0.387
Pro (%)	0.618	0.625	0.639	0.654	0.674	0.009	0.416	0.055	0.757	0.986
Arg (%)	1.036	1.078	1.083	1.085	1.076	0.120	0.760	0.352	0.357	0.789
His (%)	0.463	0.453	0.480	0.499	0.485	0.007	0.463	0.141	0.822	0.255

^a,b,c^ Means the values within a row with no common letters differ significantly (*p* < 0.05). ^1^ Lys = lysine; Phe = phenylalanine; Met = methionine; Thr = threonine; Ile = isoleucine; Leu = leucine; Val = valine; Asp = aspartic acid; Ser = serine; Glu = glutamic acid; Gly = glycine; Ala = alanine; Tyr = tyrosine; His = histidine; Arg = arginine; Pro = proline. ^2^ RA-1, diet supplemented with 100 mg/kg rosmarinic acid; RA-2, diet supplemented with 200 mg/kg rosmarinic acid; RA-3, diet supplemented with 300 mg/kg rosmarinic acid; RA-4, diet supplemented with 400 mg/kg rosmarinic acid. ^3^ Values are mean ± SEM (*n* = 8).

## Data Availability

The data presented in this study are available on request from the corresponding author. The data are not publicly available due to company policy.

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
