# Peer review of "Evaluation of Rosmarinic Acid on Broiler Growth Performance, Serum Biochemistry, Liver Antioxidant Activity, and Muscle Tissue Composition"

_animals, 2022, doi:10.3390/ani12233313_

Round 1

Reviewer 1 Report

Shang and colleagues investigated the effects of RA supplementation on broiler growth performance, serum biochemistry, meat composition, and antioxidant activity. The manuscript is well-written and well-presented. However, some minor issues need to be addressed before publication.

1.    Were the birds exposed to continuous light for 24 hours every day for 20 days? Because continuous photoperiod is typically subjected to 1-day to 7-day-old chicks, authors should double-check this.

2.    Were the chicks used for this work male or female? Please include this in section 2.1.

3.    Is there a reason why the authors ran the experiment for 20 days rather than 35/42 days? If the experiment is extended for a little longer, I believe the results will be different, especially the growth performances.

4.    Please explain in the discussion why higher doses of RA supplementation (300 and 400 mg/kg) did not produce the best results, as other research suggests otherwise.

Author Response

Response to Reviewer 1 Comments

Point 1: Were the birds exposed to continuous light for 24 hours every day for 20 days? Because continuous photoperiod is typically subjected to 1-day to 7-day-old chicks, authors should double-check this.

Response 1: Thanks for your comments. After our careful inspection, we determined that the chick house was lighted 24 h per day in this work. This is because in commercial broiler rearing, in order to pursue maximum production performance, they are raised with continuous lighting for 24 h or 23 h light:1 h darkness [1].

  1. Lewis, P.; Gous, R. J. S. A. J. o. A. S., Broilers perform better on short or step-up photoperiods. 2007, 37, (2), 90-96.

Point 2: Were the chicks used for this work male or female? Please include this in section 2.1.

Response 2: Thanks for your comments. The chicks in this work, male and female, were randomly selected for mixed rearing.

In section 2.1, lines 82-86: The sentence “One thousand one-day-old Cobb 500 white-feathered broilers (Liaocheng Fengxiang Group Co., Ltd.) with similar initial body weight (BW) (36.594 ± 0.140 g) were randomly assigned to five groups (i.e., control group and treatment group RA-1, RA-2, RA-3 and RA-4, respectively), with four replicates each group, and 50 broilers per replicate.” was changed to “One thousand one-day-old Cobb 500 male and female white-feathered broilers (Liaocheng Fengxiang Group Co., Ltd.) with similar initial body weight (BW) (36.594 ± 0.140 g) were randomly selected and assigned to five groups (i.e., control group and treatment group RA-1, RA-2, RA-3 and RA-4, respectively), with four replicates each group, and 50 broilers per replicate.”.

Point 3: Is there a reason why the authors ran the experiment for 20 days rather than 35/42 days? If the experiment is extended for a little longer, I believe the results will be different, especially the growth performances.

Response 3: Thanks for your comments. This study mainly carried out experiments on broilers at the brooding stage, so the experimental period was 20 days. Because at this time, broilers are very fragile, vulnerable to temperature, oxygen concentration and other environmental effects, and broilers are subjected to a variety of immune programs during this period, which is prone to stress reactions. Therefore, adding different doses of rosmarinic acid to the diet during this period to eliminate oxidative stress and improve its growth performance is worth studying. At the same time, the experiment of 20-40 days has also been studied. However, the main aim of the study is the effect of different doses of rosmarinic acid on the intestinal microorganisms of broilers. The experimental results and conclusions will be presented in the next article.

Point 4: Please explain in the discussion why higher doses of RA supplementation (300 and 400 mg/kg) did not produce the best results, as other research suggests otherwise.

Response 4: Thanks for your comments. High levels of RA supplementation (300 and 400 mg/kg) had no the best results on broiler growth performance in this experiment. At the same time, the literature reported that high levels of rosemary had no positive effect on the growth performance of broilers [2]. As for the reason for this result, we have not yet made further research, and there is no report in the literature. Next, we will think about and study this problem in depth. Thanks again for your comments.

  1. Ghozlan, S.; ElFar, A.; Sadek, K.; Abourawash, A.; AbdelLatif, M., Effect of Rosemary (Rosmarinus Officinalis) Dietary Supplementation in Broiler Chickens Concerning immunity, Antioxidant Status, and Performance. Alexandria Journal of Veterinary Sciences 2017, 55, (1).

Reviewer 2 Report

Few queries need to be addressed:

1. What is the rationale of conducting the feeding trial of 20 days?

2. Along with mentioning the antioxidant activity in the liver, it would be better to add the redox status data in serum and muscle as antioxidant potential of RA is focused by the authors.

3. Line 21. Replace effect with effects

4. Line 22. Replace dose rosmarinic acid with doses of rosmarinic acid.

5. Line 52-53: Sentence is repeated before in the line no. 14-15.

6. Line 74. Replace broiler reproduction with broiler production.

7. Line 183: Please remove this statement "The highest levels of TP and CLOB were observed in RA -2 group broilers" as the same results are mentioned before.

8.Immunity and lipid composition were discussed in literature but experimental data (triglycerides level in blood serum, T and B-lymphocytes assay) does not support it …either further parameters be studied or alternative literature that supports the experimental data be provided ...

9. Please label the groups RA-1 -RA-2 in the footnotes of each table.

10. For better presentation of the data it is suggested to used pooled SEM instead of individual SEM for groups.

11. The discussion needs revision. Unnecessary explanations should be replaced with the brief logical reasoning.

12. In conclusion the author mentioned "Meantime, supplementation of RA in diets can effectively improve muscle quality and flavor", please justify this statement by linking it with the results.

13. Overall the manuscript should be thoroughly reviewed again for grammatical corrections and english.

Author Response

Response to Reviewer 2 Comments

Point 1: What is the rationale of conducting the feeding trial of 20 days?

Response 1: This study mainly carried out experiments on broilers at the brooding stage, so the experimental period was 20 days. Because at this time, broilers are very fragile, vulnerable to temperature, oxygen concentration and other environmental effects, and broilers are subjected to a variety of immune programs during this period, which is prone to stress reactions. Therefore, adding different doses of rosmarinic acid to the diet during this period to eliminate oxidative stress and improve its growth performance is worth studying. At the same time, the experiment of 20-40 days has also been studied. However, the main aim of the study is the effect of different doses of rosmarinic acid on the intestinal microorganisms of broilers. The experimental results and conclusions will be presented in the next article.

Point 2: Along with mentioning the antioxidant activity in the liver, it would be better to add the redox status data in serum and muscle as antioxidant potential of RA is focused by the authors.

Response 2: Thanks for your comments. According to your suggestion, we have added serum antioxidant activity data.

The results of serum antioxidant activity are shown in Table 5. The activity of T-AOC in the serum was not significantly affected by dietary RA levels. Compared with the control group, GSH-Px, T-SOD and CAT activities were significantly increased in the RA-2 and RA-3 groups (P < 0.005) and the MDA content of the RA-2, RA-3 and RA-4 groups was significantly decreased (P < 0.005).

Item1

Control

Treatment2

SEM3

P Value

Linear

Quadratic

Cubic

RA-1

RA-2

RA-3

RA-4

T-AOC (U/mL)

8.518

9.080

9.415

9.398

9.108

0.115

0.080

0.056

0.028

0.951

GSH-Px (U/mL)

633.233b

651.663ab

657.835a

658.642a

657.745a

3.101

0.033

0.008

0.065

0.589

MDA (nmol/mL)

2.655a

2.150ab

1.836b

1.653b

1.903b

0.089

0.001

<0.001

0.007

0.611

T-SOD (U/mL)

119.710b

160.103a

168.112a

154.825a

153.345a

4.113

<0.001

0.006

<0.001

0.043

CAT (U/mL)

1.487b

1.882ab

2.128a

2.090a

1.817ab

0.748

0.036

0.075

0.007

0.854

Table 5. The effect of dietary supplementation with rosmarinic acid (RA) on serum antioxidant activity of broilers.

a,b,c Means the values within a row with no common letters differ significantly (P < 0.05). 1T-AOC=total antioxidant capacity; GSHPx=glutathione peroxidase; MDA=malonaldehyde; T-SOD=total superoxide dismutase; CAT=catalase. 2RA-1, diet supplemented with 100 mg/kg rosmarinic acid; RA-2, diet supplemented with 200 mg/kg rosmarinic acid; RA-3, diet supplemented with 300 mg/kg rosmarinic acid; RA-4, diet supplemented with 400 mg/kg rosmarinic acid. 3Values are mean ± SEM (n=8).

Point 3: Line 21. Replace effect with effects.

Response 3: Line 21: The word “effect” was changed to “effects”.

Point 4: Line 22. Replace dose rosmarinic acid with doses of rosmarinic acid.

Response 4: Line 22: The word “dose” was changed to “doses”.

Point 5: Line 52-53: Sentence is repeated before in the line no. 14-15.

Response 5: Line 14-15: The sentence “Livestock is frequently infected with pathogenic bacteria or viruses, which causes oxidative stress and multiple organ damage, thereby compromising production performance.” was deleted.

Point 6: Line 74. Replace broiler reproduction with broiler production.

Response 6: Line 74: The word “reproduction” was changed to “production”.

Point 7: Line 183: Please remove this statement "The highest levels of TP and CLOB were observed in RA -2 group broilers" as the same results are mentioned before.

Response 7: Line 183: The sentence “The highest levels of TP and CLOB were observed in RA -2 group broilers” was deleted.

Point 8: Immunity and lipid composition were discussed in literature but experimental data (triglycerides level in blood serum, T and B-lymphocytes assay) does not support it …either further parameters be studied or alternative literature that supports the experimental data be provided ...

Response 8: Thanks for your comments. According to your suggestion, we have added serum antioxidant activity data. Regarding whether adding RA to the diet can improve the Immunity function of broilers, we inferred that the immunity function of broilers could be improved based on the changes in globulin content [1, 2] and Thr content [3, 4], which are supported by the literature. The function of Immunity needs to be verified by measuring B and T lymphocytes, which is also the topic we will focus on in the next step.

  1. Grivennikov, S. I.; Greten, F. R.; Karin, M., Immunity, Inflammation, and Cancer. Cell 2010, 140, (6), 883-899.
  2. Effects of Encapsulated Rosemary Essential Oil on Growth performance, Nutrient apparent digestibility, Serum Immune and Antioxidant indexes og Weaned piglets. Chinese Journal of animal nutrition 2021, 33 (12),6470-6478
  3. Abbasi, M.; Mahdavi, A.; Samie, A.; Jahanian, R., Effects of different levels of dietary crude protein and threonine on performance, humoral immune responses and intestinal morphology of broiler chicks. Brazilian Journal of Poultry Science 2014, 16, 35-44.
  4. Montero, P.; Gimenez, B.; Perezmateos, M.; Gomezguillen, M., Oxidation stability of muscle with quercetin and rosemary during thermal and high-pressure gelation. Food Chemistry 2005, 93, (1), 17-23.

Point 9: Please label the groups RA-1 -RA-4 in the footnotes of each table.

Response 9: Thanks for your comments. Acorrding to your suggestion, we have added the notation RA-1-RA-4 to the footnotes of each table. RA-1, diet supplemented with 100 mg/kg rosmarinic acid; RA-2, diet supplemented with 200 mg/kg rosmarinic acid; RA-3, diet supplemented with 300 mg/kg rosmarinic acid; RA-4, diet sup-plemented with 400 mg/kg rosmarinic acid.

Point 10: For better presentation of the data it is suggested to used pooled SEM instead of individual SEM for groups.

Response 10: Thanks for your comments. Acorrding to your suggestion, we have used pooled SEM instead of individual SEM for groups in every tabale.

Point 11: The discussion needs revision. Unnecessary explanations should be replaced with the brief logical reasoning.

Response 11: Thanks for your comment. Based on your suggestion, we have revised and deleted some content in the Discussion section.

Lines 234-237: The sentence “The purpose of this study was to add RA to the diet and observe its effects on growth performance, serum biochemical indicators, liver antioxidant activity and muscle amino acid composition of broilers, so as to verify that RA is an alternative natural product of broiler feed antioxidants.” was deleted.

Lines 253-255: The sentence “These indices include protein composition and content, blood lipid composition and content, blood sugar content, and activities of several enzymes” was deleted.

Lines 269-273: The sentences “The liver is a crucial organ regarding the growth performance of livestock, how-ever, oxi dative stress frequently affects liver functioning, thus precluding normal growth [10]. Liver antioxidant capacity reflects the growth status of livestock, to a certain extent, and scavenging of free radicals in the liver mainly depends on various antioxidant enzymes secreted by autologous cells [33]” were changed to “The liver is a crucial organ regarding the growth performance of livestock [10]. Liver antioxidant capacity reflects the growth status of livestock, to a certain extent, and scavenging of free radicals mainly depends on various antioxidant enzymes se-creted by autologous cells [37].”.

Point 12: In conclusion the author mentioned "Meantime, supplementation of RA in diets can effectively improve muscle quality and flavor", please justify this statement by linking it with the results.

Response 12: Thanks for your comments. According to your suggestion, we have modified the relevant content to link it with the results

Line 319-320: The sentence “Meantime, supplementation of RA in diets can effectively improve mus-

cle quality and flavor.” was changed to “Meantime, supplementation of RA in diets could increase the content of essential AAs (Lys, Thr) and flavor AAs (Glu, Ser), so as to improve muscle quality and flavor.”.

Point 13: Overall the manuscript should be thoroughly reviewed again for grammatical corrections and english.

Response 13: According to your suggestion, we have entrusted Editage (www.editage.cn) to polish and edit the English language of full manuscript.

Reviewer 3 Report

Work "Evaluation of Rosmarinic Acid on Broiler Growth Performance, Serum Biochemistry, Liver Antioxidant Activity, and Muscle Tissue Composition" is interesting on topical issues. In the reviewer's opinion, however, the article requires correction before publication.

Introduction

Line 68-70. The reviewer disagrees with this statement. A lot of research has already been done in this direction using RA. So I am asking the authors to analyze the research results to date (in the Introduction section) and then refer them to their results in the Discussion section.

Materials and methods

This section is discussed in detail; there is no doubt about the structure of the experiment and its course.

In this part, I propose to add that the authors use SEM (explain the abbreviation).

Lines 86-87. The authors report: "RA solid powder (Hunan Jinhan Pharmaceutical Co. Ltd.) was added to the basal diet at 0, 100, 200, 300, or 400 mg/kg." Why were such quantities selected? Based on literature data, or maybe the results of preliminary research?

Tables 2, 3, 4. I propose to standardize the assignment of letters with average values. Sometimes the letter "a" is at the lowest value, other times at the highest. I propose to add the number of attempts under the table based on which the average was calculated.

Discussion

Lines 234-237. This fragment should not be found here; please move it to Introduction or delete it.

The test results must be thoroughly discussed in the literature (see comment above).

Author Response

Response to Reviewer 3 Comments

Point 1: Introduction

Line 68-70. The reviewer disagrees with this statement. A lot of research has already been done in this direction using RA. So I am asking the authors to analyze the research results to date (in the Introduction section) and then refer them to their results in the Discussion section.

Response 1: Thanks for your comments. Acorrding to your opinion, we have made changes to the relevant content.

Line 68-70: The sentences “However, there are few studies on its use as a feed additive in animal rearing, and its effects in poultry rearing are so far unclear, especially whether it has antioxidant ac-tivity in broiler rearing.” were changed to “In recent years, the application of rosemary extract or rosemary essential oil as an ad-ditive in livestock and poultry rearing was a hot research topic. Norouzi et al. reported the effect of different dietary levels of rosemary herb powders on the growth perfor-mance, carcass traits and ileal micro-biota of broilers [17]. Ghozlan et al. study showed that adding high concentrations of rosemary to the diet had no significant effect on the growth performance of broilers, and appropriate concentrations of rosemary increased the immunity and antioxidant activities of broilers [18]. The addition of rosemary es-sential oils improved some production performance and antioxidant enzyme activity (glutathione peroxidase) in heat-stressed laying hens [19]. Dietary supplementation with rosemary (0.5%) was more effective in improving aflatoxin B1-contaminated ti-lapia feed and enhancing fish immunity [20]. However, studies on the antioxidant, an-ti-inflammatory, and antiviral properties of RA as the main component of rosemary had mainly focused on rats [21-25]. Whether RA as the main component of rosemary added to the diet has the above functions in livestock and poultry rearing deserves further study.”.

Point 2: Materials and methods

This section is discussed in detail; there is no doubt about the structure of the experiment and its course.

In this part, I propose to add that the authors use SEM (explain the abbreviation).

Response 2: Thanks for your comments. Acorrding to your opinion, we have made changes to the relevant content.

Line 158: The sentence “Statistical significance is reported at P < 0.05.” was changed to “Values are expressed as means ± SEM (Standard error of mean) and statistical signifi-cance is reported at P < 0.05.”.

Point 3: Lines 86-87. The authors report: "RA solid powder (Hunan Jinhan Pharmaceutical Co. Ltd.) was added to the basal diet at 0, 100, 200, 300, or 400 mg/kg." Why were such quantities selected? Based on literature data, or maybe the results of preliminary research?

Response 3: Thanks for your comments. According to the results of preliminary research, it was found that dietary supplementation of high concentrations of rosmarinic acid (500 mg/kg) did not significantly improve the growth performance of broilers. At the same time, the literature reported that high levels of rosemary had no positive effect on the growth performance of broilers [1]. Therefore, we selected 0, 100, 200, 300 and 400 mg/kg of rosmarinic acid as an additive in the diet for experimental studies.

  1. Ghozlan, S.; ElFar, A.; Sadek, K.; Abourawash, A.; AbdelLatif, M., Effect of Rosemary (Rosmarinus Officinalis) Dietary Supplementation in Broiler Chickens Concerning immunity, Antioxidant Status, and Performance. Alexandria Journal of Veterinary Sciences 2017, 55, (1).

Point 4: Tables 2, 3, 4. I propose to standardize the assignment of letters with average values. Sometimes the letter "a" is at the lowest value, other times at the highest. I propose to add the number of attempts under the table based on which the average was calculated.

Response 4: Thanks for your comments. Acorrding to your opinion, we have added the number of attempts under the every table based on which the average was calculated.

For example in the footnote of Table 2, the sentence “3Values are mean ± SEM (n=4).” was added. The value of "n" is the number of attempts.

Point 5: Discussion

Lines 234-237. This fragment should not be found here; please move it to Introduction or delete it.

Response 5: Lines 234-237: The sentence “The purpose of this study was to add RA to the diet and observe its effects on growth performance, serum biochemical indicators, liver antioxidant activity and muscle amino acid composition of broilers, so as to verify that RA is an alternative natural product of broiler feed antioxidants.” was deleted.

Point 6: The test results must be thoroughly discussed in the literature (see comment above).

Response 6: Thanks for your comment. Based on your suggestion, we have revised and deleted some content in the Discussion section.

Lines 234-237: The sentence “The purpose of this study was to add RA to the diet and observe its effects on growth performance, serum biochemical indicators, liver antioxidant activity and muscle amino acid composition of broilers, so as to verify that RA is an alternative natural product of broiler feed antioxidants.” was deleted.

Lines 253-255: The sentence “These indices include protein composition and content, blood lipid composition and content, blood sugar content, and activities of several enzymes” was deleted.

Lines 269-273: The sentences “The liver is a crucial organ regarding the growth performance of livestock, how-ever, oxi dative stress frequently affects liver functioning, thus precluding normal growth [10]. Liver antioxidant capacity reflects the growth status of livestock, to a certain extent, and scavenging of free radicals in the liver mainly depends on various antioxidant enzymes secreted by autologous cells [33]” were changed to “The liver is a crucial organ regarding the growth performance of livestock [10]. Liver antioxidant capacity reflects the growth status of livestock, to a certain extent, and scavenging of free radicals mainly depends on various antioxidant enzymes se-creted by autologous cells [39].”.

Lines 284-285: The sentence “MDA content was significantly decreased, and antioxidant capacity of T-AOC was significantly increased in this study.”was changed to “MDA content of liver and serum was significantly decreased in this study.”.

Round 2

Reviewer 2 Report

The revised manuscript is an improved version now.

Only one suggestion is if this trial assesses RA's effects on growth performance during the "starter period", then this could be mentioned in the title.

Reviewer 3 Report

The reviewer thanks the authors for the changes and clarifications.

The authors' answers are not in doubt.

The work was improved following the reviewer's suggestions.

The introduced changes significantly improved the quality of manuscript.